# Evaluation of Hydrodynamic and Thermal Behaviour of Non-Newtonian-Nanofluid Mixing in a Chaotic Micromixer

**DOI:** 10.3390/mi13060933

**Published:** 2022-06-11

**Authors:** Naas Toufik Tayeb, Shakhawat Hossain, Abid Hossain Khan, Telha Mostefa, Kwang-Yong Kim

**Affiliations:** 1Gas Turbine Joint Research Team, University of Djelfa, Djelfa 17000, Algeria; toufiknaas@gmail.com; 2Department of Industrial and Production Engineering, Jashore University of Science and Technology, Jashore 7408, Bangladesh; 3Institute of Nuclear Power Engineering, Bangladesh University of Engineering and Technology, Dhaka 1000, Bangladesh; khanabidhossain@gmail.com; 4Mechanical Engineering Department, Ziane Achour University of Djelfa, Djelfa 17000, Algeria; telhamostefa@gmail.com; 5Department of Mechanical Engineering, Inha University, 100 Inha-ro, Michuhol-gu, Incheon 22212, Korea

**Keywords:** chaotic micromixer, Nano-Non-Newtonian fluid, mass mixing index, thermal mixing index, low generalized Reynolds number, minimal mixing energy cost

## Abstract

Three-dimensional numerical investigations of a novel passive micromixer were carried out to analyze the hydrodynamic and thermal behaviors of Nano-Non-Newtonian fluids. Mass and heat transfer characteristics of two heated fluids have been investigated to understand the quantitative and qualitative fluid faction distributions with temperature homogenization. The effect of fluid behavior and different Al_2_O_3_ nanoparticles concentrations on the pressure drop and thermal mixing performances were studied for different Reynolds number (from 0.1 to 25). The performance improvement simulation was conducted in intervals of various Nanoparticles concentrations (φ = 0 to 5%) with Power-law index (n) using CFD. The proposed micromixer displayed a mixing energy cost of 50–60 comparable to that achieved for a recent micromixer (2021y) in terms of fluid homogenization. The analysis exhibited that for high nanofluid concentrations, having a strong chaotic flow enhances significantly the hydrodynamic and thermal performances for all Reynolds numbers. The visualization of vortex core region of mass fraction and path lines presents that the proposed design exhibits a rapid thermal mixing rate that tends to 0.99%, and a mass fraction mixing rate of more than 0.93% with very low pressure losses, thus the proposed micromixer can be utilized to enhance homogenization in different Nano-Non-Newtonian mechanism with minimum energy.

## 1. Introduction

Heat and mass transfer produced by different fluid concentrations is generally applied in high mixing processes that affect the dynamic transport of chemical species within physical chaotic advection. Mixing of non-Newtonian fluids process of micromixers occurs in multiple applications of various devices and industrial applications that have significant utility in bioengineering fields [1], chemical engineering [2], extraction, polymerization techniques and cleaning systems [3,4,5,6,7], etc. To enhance the kinematic behaviour [8] and fluid homogenization performance, many researchers contribute various solutions using chaotic advection inside microchannels [9,10] while reducing pressure drops [11,12,13,14].

Xia et al. [15] investigated experimentally and numerically different passive micromixers with multi-layer crossing channels. Their proposed micromixer can give a fast mixing (more than 95%) at minimum Reynolds number (not exceeding 40), for example, a high mixing index of 0.96 was found for very low flow regimes. The same geometry was selected by Hossain et al. [16] to carry out a parametric study. To develop the mixing efficiency, the effect of the geometrical parameter was analyzed at (not exceeding 40) low Reynolds numbers.

The mixing enhancement of split and recombination micromixers using Newtonian fluid flow with low regimes was investigated numerically by Hossain et al. [17]. They evaluated the homogenization performance and the pressure-drop characteristics compared to the TLCCM micromixer which was selected by Xia et al. [15], in a range of Reynolds numbers between 0.2 and 120. Raza et al. [18,19] developed the same problem to create an unbalanced split and recombination micromixer (1.5 mm). They measured numerically the mixing efficiency within various cross-sections via low flow regime. They found that the optimal configuration has 88% mixing efficiency. A parametric study using five geometric parameters of the micromixer was performed by Naas et al. [8]. They presented the effect of a low Reynolds number on mixing efficiency. Their proposed short micromixer gives mixing rates greater than 99% throughout the Reynolds number range in a short length of three-dimensional micromixer, which is rarely achieved in early micromixer configurations.

The effects of thermal condition of non-Newtonian fluids rheology were investigated by Naas et al. for C-shape chaotic geometry [20,21] and Multi-layer micromixers [22]. They selected a highly efficient heat mixing system and better thermodynamic performance, for rheology index ranging from 1 to 0.49. The results showed that the decrease in heat transfer flow of shear-thinning flows was higher than that of Newtonian fluids.

In order to enhance the heat transfer efficiency, several researchers carried out numerically [23,24,25,26,27,28,29,30] and experimentally [31,32] the thermo-physical enhancement of various applied thermal engineering systems. They used effective nanoparticles such as: Al_2_O_3_ [23,24], H2O/SWCNT [29], CuO [28,29,30]. The authors proved experimentally that the nanofluids have significantly more powerful thermal conductivities than the same flow with zero nano concentration.

Inside tubes, Xuan and Li [33] investigated experimentally the thermal performance of nanofluid swirling under wall heat transfer. Compared with pure water, the heat transfer coefficient was augmented more than 39% with the Cu nanoparticles concentration of 2.0%.

Within rectangular microchannels, convection laminar regimes with nano-shear thinning flow have been carried out by Esmaeilnejad et al. [34]. Their obtained results presented that, with Peclet number of 700 and 4% particle concentration, the thermal coefficient will decease around 27.2% and the pressure drop will increase roughly 50.7%. Evangelos et al. [35] studied the mixing performances of heated water with the effect of an electromagnetic field. For different mixers, Pouya [36] investigated numerically, the heat transfer enhancement and mixing quality of a hybrid nanofluid. They found that for high Reynolds numbers and a constant frequency, the mixing rate increased over time.

In other works [37], the authors created a chaotic geometry called C-shape to enhance the heat transfer characteristics of non-Newtonian nanofluid. They found that the development of chaotic advection causes more enhancement to thermal rate in comparison with pumping power.

In recent years, Antar and Kacem [24] analyzed the enhancement of forced convection flow within a straight pipe under the effect of nano-particles of Al_2_O_3_subjected to a fixed wall of heat. They examined the effects of both fluid behavior index and nanofluid concentrations on the gradient temperature and heat transfer rates. Their obtained results indicated that, for various Peclet and Brinkman numbers, the heat transfer rate increases when nanoparticles concentration.

Due to the diminutive measurement of micromixers, it’s difficult to enhance the molecular diffusion operation. While the improvement in dynamic behaviors is limited, the injection of nanofluid concentrations is a suitable manner in which to enhance mass transfer and thermal mixing efficiency within a passive micromixer. Therefore, our contribution aims to improve high mixing performances for Nano-non-Newtonian fluids in terms of mass and heat transfer inside the optimum multi-layer micromixer as it was recently used by Naas et al. [8] beneath a very low Reynolds number. Various nanoparticles concentrations with different values of fluid behavior index were proposed to investigate the chaotic flow formation and thermal mixing performances within the suggested micromixer. In order to get important energy efficiency, the homogenization of the fluid indexes and mixing energy cost will be appraised.

## 2. Micromixer Structure and Problem Statement

A novel micromixer was suggested in this study, namely a Two-Layer Crossing Modified geometry (TLCM) which was used recently and firstly by Naas et al. [8] to achieve High-Mixing Processing under the effect of nanofluid concentrations for power-law non-Newtonian fluid [24], see Figure 1. The micromixer is constructed of a couple of twisted channels; the higher and lower channels are arranged with a periodic chamber. The following mixing components occur in reconstructed forms of several grooves. Dimensions detail is presented in Table 1, where D is the chamber diameter, d is the grooves diameter, d_hyd_ is the hydraulic diameter, l is the distance between the inlets and L* is the length of the geometry. The same nano-non-Newtonian fluid of Antar and Kacem [23,24] and Santra et al. [36] was proposed as working fluid, represented in Table 2 and Table 3 respectively.

Heated and cold Nano-non-Newtonian fluids flow from the inlets with a uniform velocity profile. The outsides are deemed adiabatic and the other boundaries include no-slip. Pressure outlet condition is awarded at the outflow section. A numerical CFD code Fluent© numerically solved the governing equations, which are given by the following expressions [22,38]:(1)$$div\vec{V}=0$$
where $\;\vec{V}$ is the velocity vector.
(2)$$\vec{V}\cdot \bar{\bar{\nabla }}\vec{V}\;=\;-\frac{1}{\rho_{nf}}\vec{\nabla }P+div\tau$$
where σ (Pa) is shear stress and *P* is the pressure.
(3)$$\rho_{nf}c_{nf}\vec{V}\cdot \vec{\nabla }T=\;\;\lambda_{nf}\Delta T$$
where $\rho_{nf}$, $\lambda_{nf}$ and T are the density, the conductivity and the temperature of the nanofluid, respectively.

The constitutive connection among the shear rate $\dot{\gamma }$ (s^−1^) and shear stress τ (Pa) can be characterized by a simple power-law equation:(4)$$\tau \;=\;m{\dot{\gamma }}^{n}$$
where, *m* (Pas^−1^) is fluid consistency index and n is the fluid behavior index.

The apparent viscosity is:(5)$$\mu_{nf}\;=\;k{\dot{\gamma }}^{n-1}$$

The advised boundary conditions are:-Uniform velocity profile imposed to the inlets flow, and the temperatures equal to *T*_min_ = 300 k for one inlet, and the other one *T*_max_ = 330 k-No-slip conditions within the solid walls.-Pressure outlet condition is considered at the outlet section flow.

### 2.1. Mass Transfer Characteristics of the Chaotic Flows

Metzner and Reed [39,40] defined the generalized Reynolds number (Re_g_) for power-law fluids as follows:(6)Reg=ρnfu2−ndhydn[8n−1(b∗+a∗n)nm]

$a^{*}$ Geometric parameter ($a^{*}$ = 0.2121)$b^{*}$ Geometric parameter ($b^{*}$ = 0.6771)$\rho_{\mathrm{nf}}$ Density $\left[\mathrm{kg}\cdot m^{-3}\right]$u Average speed $\left[m\cdot s^{-1}\right]$$d_{\mathrm{hyd}}$ Hydraulic diameter [m]m Consistency index (N s^n^ m^−2^)n Rheological behavior index of the fluid

Mixing efficiency of two Nano-non-Newtonian fluids inside the micromixer is characterized by:(7)$${\mathrm{MM}}_{i}=1-\frac{\sigma }{\sigma_{0\;}}$$

σ  is the standard deviation measured by post-CFD.$\sigma_{0}$ is the maximum standard deviation.



(8)
$$\sigma^{2}=\frac{1}{N}\overset{N}{\underset{i=1}{\sum }}{\left(C_{i}-\bar{C}\right)}^{2}$$



For fully mixed fluids, the standard deviation rate was taken from the minimum values and maximum for unmixed fluids. N presents the total number of nodes. $\bar{C}$ is the average mass fraction. The maximum standard deviation $\sigma^{2}$ is measured by:(9)$$\sigma_{0}^{2}=\left(1-\bar{C}\right)$$

The mixing energy cost (MEC) is used to decide the effectiveness of the micromixer and is established by merging the pressure losses and the mixing degree, as follows [41]:(10)$$\mathrm{MEC}=\frac{\Delta P\times Q}{\mathrm{MMi}}$$
where ΔP and Q are the pressure drop and the flow rate (m^3^/s) along the geometry, respectively.

### 2.2. Thermal Characteristics of the Chaotic Flows

Heat transfer coefficient, h, different inlet temperature is assumed as:(11)$$h=\frac{q\prime \prime }{\left(T_{b}-T_{w}\right)}$$
where, T_b_ (k) is the mean bulk temperature fluid, T_w_ (k) is the perimeter average wall temperature and q″ (w/m^2^) is the wall heat flux.

These two temperatures are defined as:(12)$${\;T}_{w}\left(s\right)=\frac{1}{P}\int_{P}T_{w}\mathrm{dp}$$
(13)$$T_{b}\left(s\right)=\frac{1}{{\mathrm{AU}}_{i}}\overset{}{\underset{A}{\iint }}\vec{V}\cdot \vec{n}T\cdot \mathrm{dA}$$

The mean heat transfer coefficient, $h_{\mathrm{mean}}$, is defined as:(14)$$h_{\mathrm{mean}}=\frac{1}{L}\overset{L}{\underset{0}{\int }}h\left(s\right)\mathrm{ds}$$

The Thermal Mixing index (${\mathrm{TM}}_{i}$) utilized by Naas et al. [22] of two fluids (hot and cold) is given by the following equation:(15)$${\mathrm{TM}}_{i}=1-\frac{\iint_{s}^{}\left|T-T_{\mathrm{avg}}\right|\mathrm{ds}}{\left(T_{\max}-T_{\min}\right)s}$$
where, T_avg_ = (T_min_ + T_max_)/2. $T_{\min}$= 300 K and $T_{\max}=330$.

K The values of ${\mathrm{TM}}_{i}$ range from 0 for the unmixed flow case, to 1 for fully mixed flow.

In order to confirm the efficiency of the scalar temperature T which was measured between two values; (T_a_, T_b_) at a given section, a probability density function PDF % (T) was calculated which was equal to the number of nodes within (T_a_, T_b_) divided by the total number of nodes [22].

All governing equations in the present work were solved in a laminar flow by using ANSYS Fluent 16^©^ CFD software [42], which works by the (FVM) finite volumes method. The SIMPLEC scheme was chosen for pressure and velocity coupling. To determine the mass and momentum equations a second-order upwind scheme was selected. The computations were ensured and simulated to be converged at 10^−7^ of root mean square (RMS) residual values. Non-Newtonian power-law fluids were done as a working fluid for various Al_2_O_3_ nanoparticles concentrations.

## 3. Mesh Sensitivity Test

To check the sensitivity of the CFD results, a quantitative grid study was carried outby varying the total number of cells. Using an unstructured mesh with uniform tetrahedral cells, four mesh grids were analyzed ranging from 100,000 to 800,000.

The evolutions of the velocity along the X-axis of the outlet section with a Reynolds number equal to 10 are shown in Figure 2. It can be observed that the outlet velocity rates were sensitive to the mesh, except for the mesh densities with 700,000 and 800,000 nodes where no important difference is seen. As a consequence, the 700,000 nodes grid is selected as the favorable mesh for the analyses (Figure 2 and Figure 3).

## 4. Results and Discussion

Kinematic and thermal behaviors of mixing Nano-non-Newtonian flows are investigated in detail within novel micromixers, which are compared with potential micromixers used recently in the literature. Heat, mass transfer and fluid mixing process are investigated for several low generalized Reynolds number ranging between 0.1 to 25.

### 4.1. CDF Validation Case

The numerical solution procedure of mass transfer performances has been reported and validated thoroughly by comparing the present results with the results of Xia et al. [15] and Hossain et al. [16], for Reynolds number equal to 0.2, quantitative and quantitative comparison for mixing efficiency index and mass transfer contours for different successive cross-sectional planes, as shown in Figure 4.

An excellent agreement is seen to exist between the present numerical values and the literature values of mass mixing index. Based on these comparisons, it is perhaps reasonable to conclude that the present results are reliable to within ±0.4%. Deviations of this order are not at all uncommon in numerical studies and arise due to the differences in the flow schematics, problem formulations, grid and/or domain sizes, discretization schemes and numerical methods.

Moreover, a quantitative numerical validation was carried out with those obtained by Ning et al. (2016) [43], and the results present a heat transfer rate as a function of various Reynolds numbers for non-Newtonian cases. The comparison is satisfactory and revealed good agreements among results which are shown in Figure 5.

In addition, we added another validation with those obtained by Jibo et al. [44]. The problem was analyzed for multiphase fluids of mixing enhancement inside micromixers, see Figure 6, where the relative error with the numerical results is less than 1%.

### 4.2. Mass Transfer and Fluid Mixing Processing

Flow visualization of mass fraction between two nanofluids is shown in Figure 7. Different cases of fluid concentrations are subjected within the fluid pattern to understand the development of visual mixing inside the new configuration. We remark that the fluid mixing augments with the Reg for all fluid behavior index n and there is no exactly black region in the second unites. Therefore, the new micromixers have a quick mass transfer performance.

Figure 8 shows the evaluation of the mass mixing efficiency of two fluids as a function of Re_g_ for various cases of nanofluid concentrations (φ = 0.5 to 5%). As the flow homogenization was only performed by the nanoparticles, the prevalence of Reynolds number is very low (Re < 5) and the flow behaviors were not effective for improving mixing (molecular diffusion dominates). When the generalized Reynolds fluid number increased, the homogenization was more effective, and the mixing intensity developed rapidly. Note that in large Reynolds numbers, the nano-particles are more effective, and the mixing intensity develops quickly so the most select mixing state is reached when the concentration increases. Moreover, it is observed that the proposed micromixer shows a 2.22% enhancement of mixing intensity when the fluid behavior index decreases to 0.46, as compared to the case of n = 0.88.

The homogeneity achieved over midlines at the exit of the several cases of nano-fluid concentrations inside the micromixer at a given Re_g_ is demonstrated by the standard deviation like mass transfer profiles as shown in Figure 5. Significant changes in the values of the mass transfer for all cases of Nano-non-Newtonian fluids can be observed. For the case of φ = 5%, the results indicates that the best degree of mixing done at the exit of the micromixer.

To explain the structure of the fluid flow inside the proposed micromixer, a visualization of vectors and streamlines of the mass fraction and vortex core regions are shown in Figure 9. The kinematic behavior has an important role in enhancing homogenization for nanofluids. As can be seen in Figure 10, the micromixer has a single strong vortex region inside each corner which develops the mixing rate inwardly of the geometry, whereas it has a low-pressure drop near the outlet part. Furthermore, we can see that the flow is more chaotic and dynamic due to the structure and the curve of the configuration. Moreover, it is remarkable that the pathline inside the selected new micromixer produces a reversed flow pattern and strong secondary flows are created which can enhance more mass transfer efficiency and ensure excellent quality of homogenization.

### 4.3. Heat Transfer and Thermal Mixing Processing

Thermal mixing fluids between hot and cold Nano-non Newtonian fluids are measured for various cases of nanofluid concentrations (φ = 0.5 to 5%). For this, the cold fluid is injected in one inlet at 300K and the heated fluid is injected in the other inlet at 330 K, see Figure 11. In this part, we investigate thermal mixing performances and mining energy coast in the present micromixer, and its performances will be compared to other strong micromixers. Figure 11 shows a top view of the temperature contours in three cases of nanofluid concentration (φ = 0, 2.5 and 5%) for Reg ranging from 0.1 to 25. For a given value of nanofluid concentration or fluid behavior index, the thermal homogenization quality is more vigorous when the Re_g_ is more important. The development of the generalized number highly improves the dynamic of the movement, the kinematic of the fluid nanoparticles varies considerably and the mixing intensity will be improved.

The influence of the nanofluid concentration loading on the improved thermal efficiency is analyzed, see Figure 12, Figure 13 and Figure 14. Figure 12 shows the evolution of heat transfer coefficient for several generalized Reynolds numbers with different nano-fluid concentrations (φ = 0.5 to 5%). As the nanofluid concentration ranges from 0.5% to 5%, the heat transfer augments from 122 W/m^2^ K to 222 W/m^2^ K for Re_g_ = 25. It can be resolved that other devices besides thermal conductivity increase can be efficient for thermal flow enhancement. Thermal nanofluid performance is under the effect of various parameters. It appears that nanofluid behavior may develop the temperature gradient within the flow and thus increase the heat of the nanofluid, see Figure 13. Figure 14 shows the evaluation of the thermal mixing efficiency of two fluids for different Re_g_ for various cases of nanofluid concentrations (φ=0.5 to 5%).

Due to the heat transfer being done only by conduction, it is indicated that the mixing quality is not affected by the fluid concentrations for low Re_g_ numbers. Therefore, the rheology fluid index n has no effect on the mode of energy transfer in the flow. When Re_g_ augments, the flow is chaotic which leads to greater mixing. This behavior enhances greatly the thermal efficiency as a function of Re_g_ number for all cases of nanofluid concentrations. For Re_g_ numbers equal to 25, the mixing degree is close to 1 and the quality of mixing is perfect. The degree of thermal mixing has an insignificant difference as a function of the fluid concentrations considered, so the fluid behavior index n has the same effect on the TMI as described above.

Table 4 presents the probability density function of temperature distribution for different cross-sections. As it is known, the greatest PDF is 100% for T = 315 K. Despite the fluid passing through the micromixer, the fluids are well mixed and tend to be homogenized (T= 315 K) under the effect of the chaotic behavior of the flow. For all cases of various nanofluid concentrations, the temperature distribution from the third plane (P3) is examined in a small variety of the order of two Kelvin, where the best of this arrangement is compared to the coveted mixing fluid temperature, 315 K.

To choose the better Nano-non-Newtonian fluid concentration with high mixing efficiency and lower cost of mixing energy, Figure 15 presents Mixing Energy Cost under the effect of generalized Reynolds number ranging (Re_g_ = 0.1 to 25) for various nanofluid concentrations (φ = 0.5 to 5%). The flow of low Re_g_ has qualitatively the same behavior in terms of cost of the mixing energy, due to the fluid being more viscous and the secondary flows not yet active. When the Re_g_ overrun the value of 5, the difference enhances obviously for both the kinematic and thermal process. As can be seen, when the Reg increases, the MEC become larger inside the micromixers because the fluid flow turns out more sheared and agitated. Moreover, the fluid flow concentration of 0.5% displays low rates of energy cost compared to the other cases.

The development of Mixing Energy Costis illustrated in Table 5 for various Reynolds numbers. For cases of Reynolds numbers, it can be clearly remarked that the present micromixer has the most economical mixing energy cost compared to the micromixer studied recently by Embarek et al. in [39]. In addition, to compare quantitatively the mixing index, pressure-losses, and the mixing energy cost, Table 6 shows different recent works (2017, 2019, 2021y) for these parameters at fixed Reynolds number (Re = 70). It is shown that the proposed micromixer has the lower pressure drops and the higher mixing index value of 99.99%, and the best mixing energy cost.

## 5. Conclusions

A numerical CFD work was performed to analyze the flow characteristics of Nano-non-Newtonian shear-thinning fluid inside a novel micromixer. This study outlines the hydrodynamic and thermal mixing performances for different generalized Reynolds numbers with nanofluid concentrations (φ = 0.5 to 5%). According to this study, the following points can be given:-Effects of generalized Reynolds numbers on the hydrodynamic behavior of non-Newtonian flow were improved within the proposed micromixers.-Strong secondary flows are created inside the micromixer to enhance the mixing quality for all cases of nanofluid concentration.-The non-Newtonian fluid of φ = 0.5% exhibits low mass transfer compared to the preferable nanofluid concentration case (φ = 5%).-As Reynolds number increases, the flow visualization of both heat and mass transfer revealed that the vortex created in the micromixer had more vigorous intensity.-Higher rates of generalized Reynolds number have more effects to increase both mass and thermal homogenization rates.-The heat transfer coefficient increases from 122 W/m^2^K to 222 W/m^2^K when φ rises from 0.5 to 5%.-The cases of low fluid behavior index n have a more effective improvement in the mixing efficiency than the other cases-For all of Reg, high intensity thermal and fluid mixing is obtained for high nanofluid concentration (φ = 5%).

## Figures and Tables

**Figure 1 micromachines-13-00933-f001:**
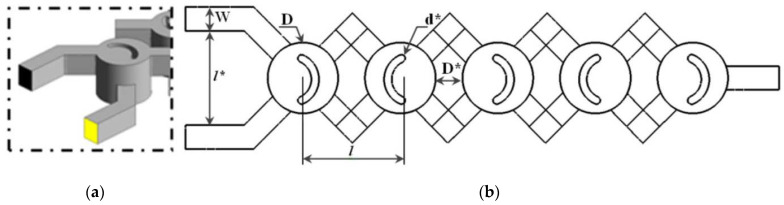
(**a**): 3-D view design of the proposed micromixers, (**b**): geometric parameters.

**Figure 2 micromachines-13-00933-f002:**
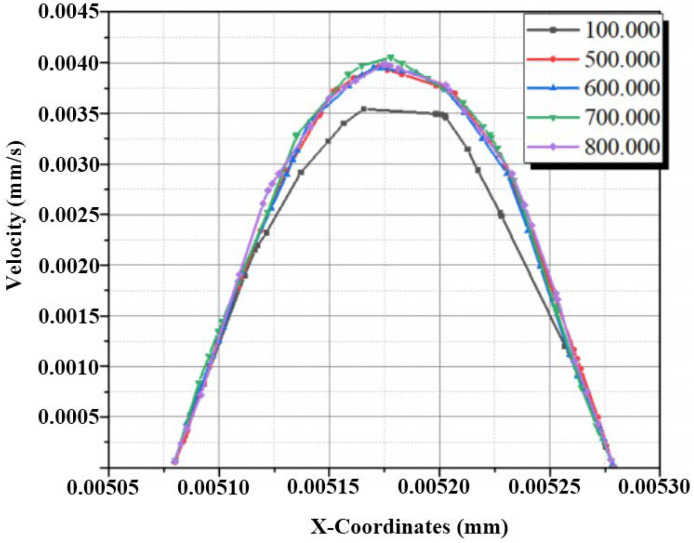
Evolutions of outlet velocities at the exit mid-line of X coordinates.

**Figure 3 micromachines-13-00933-f003:**
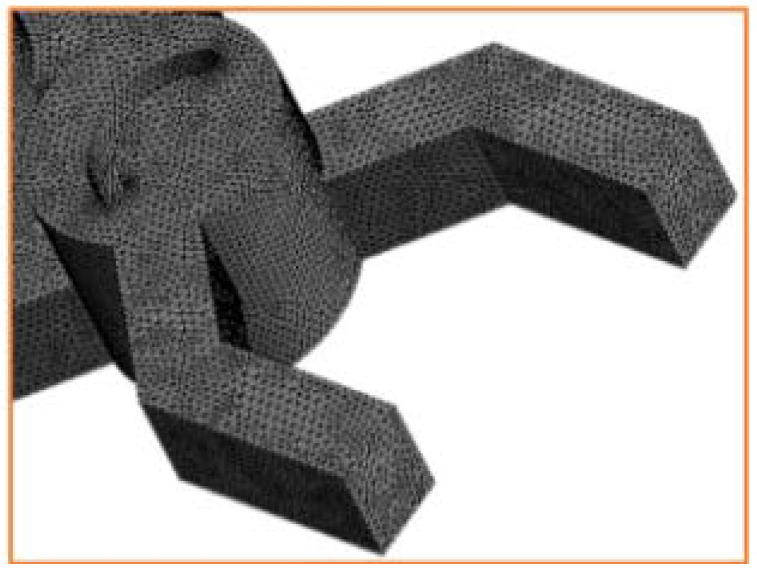
Structure of the generated mesh grid.

**Figure 4 micromachines-13-00933-f004:**
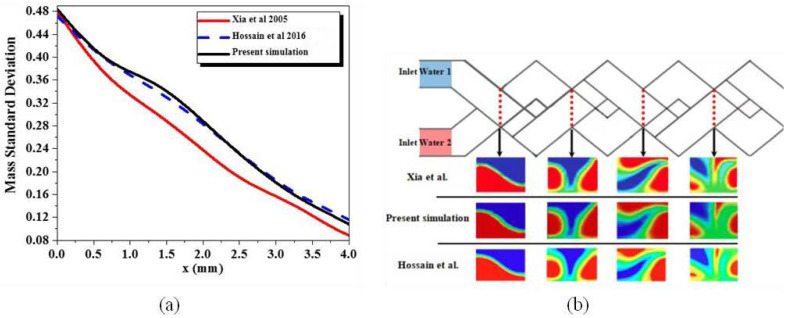
Quantitative validation for mixing efficiency index (**a**), and quantitative validation of local mass fraction contours for various planes (**b**) With the results of Xia et al., reproduced with permission from [15] and Hossain et al. reproduced with permission from [16].

**Figure 5 micromachines-13-00933-f005:**
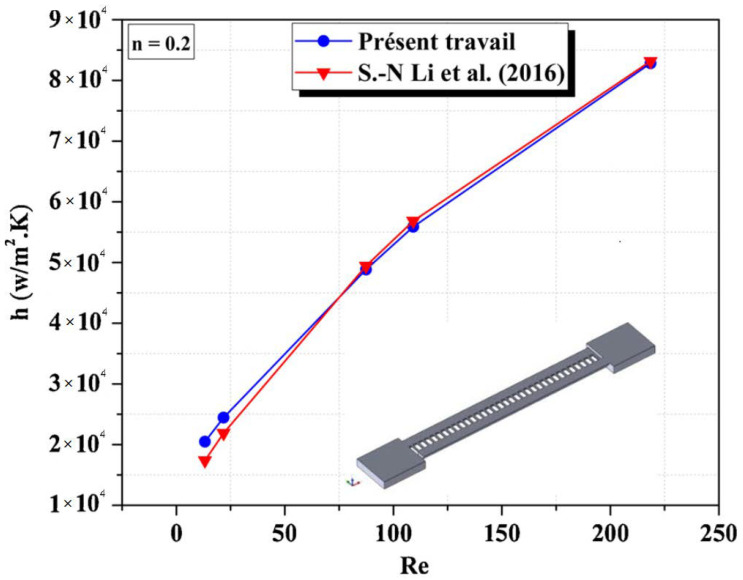
Heat transfer coefficient Vs Reynolds number for non-Newtonian case with Li et al. reproduced with permission from [43].

**Figure 6 micromachines-13-00933-f006:**
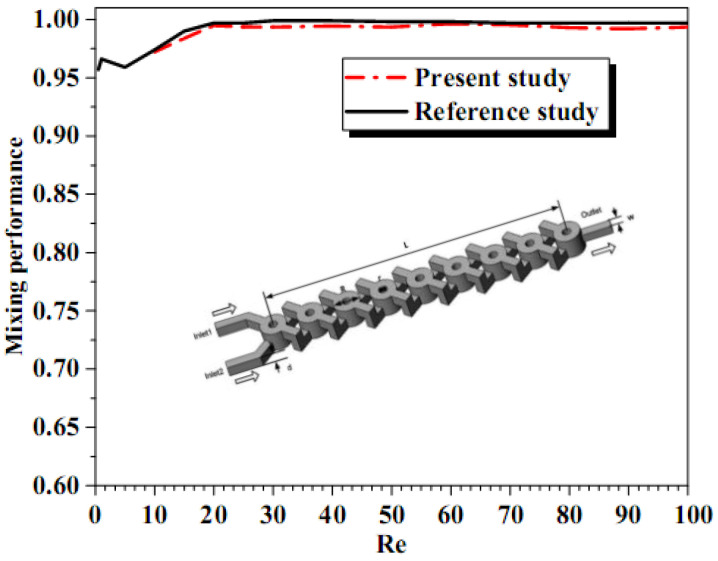
Evolution of mixing rates Vs Reynolds number with Jibo et al. [44].

**Figure 7 micromachines-13-00933-f007:**
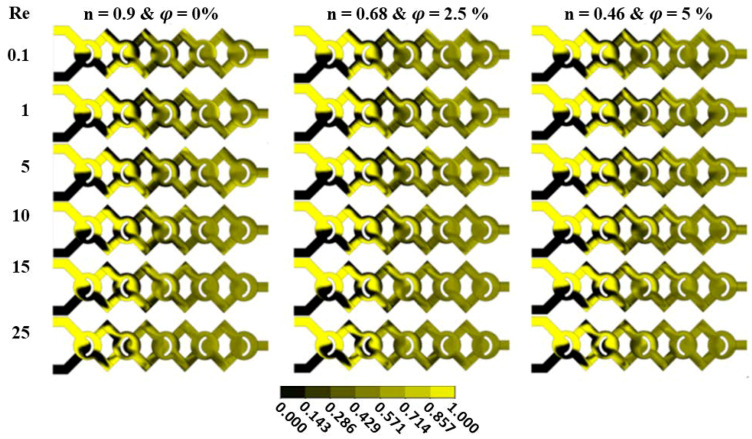
Mass fraction contours at various Reynolds numbers with different fluid concentrations (φ = 0.5 to 5%).

**Figure 8 micromachines-13-00933-f008:**
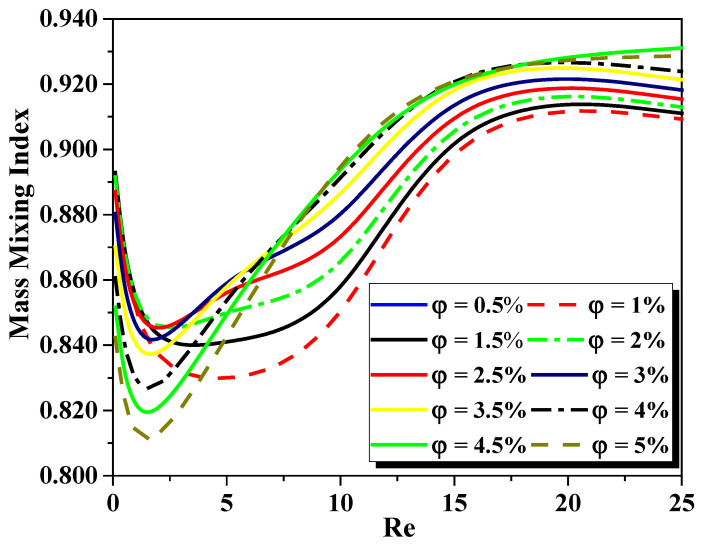
Development of mass mixing performance for different Reynolds numbers with variation cases of nanofluid concentration (φ = 0.5 to 5%).

**Figure 9 micromachines-13-00933-f009:**
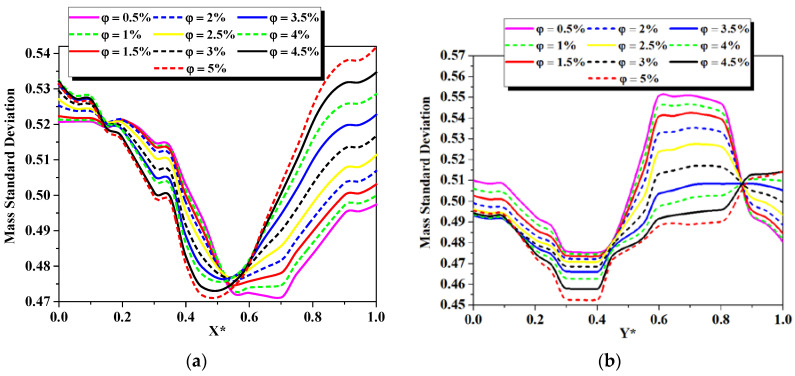
Evolution of the mass standard deviation along the (**a**): exit X-line and (**b**) exit Y-line of different cases of nano-fluid concentration (φ = 0.5 to 5%) at fixed Re_g_ = 25.

**Figure 10 micromachines-13-00933-f010:**
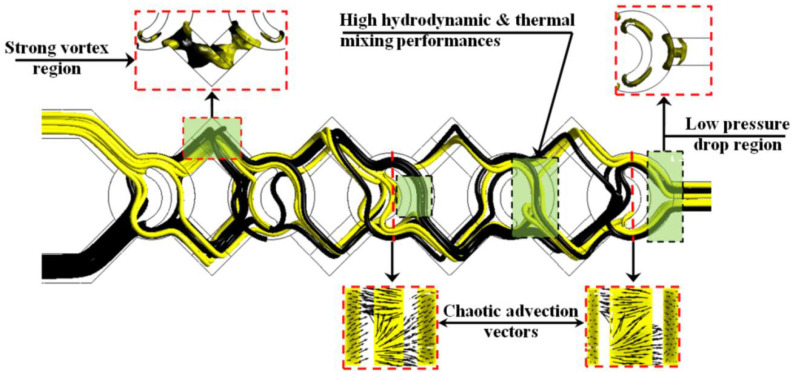
Streamlines and vectors of the mass fraction with vortex core region.

**Figure 11 micromachines-13-00933-f011:**
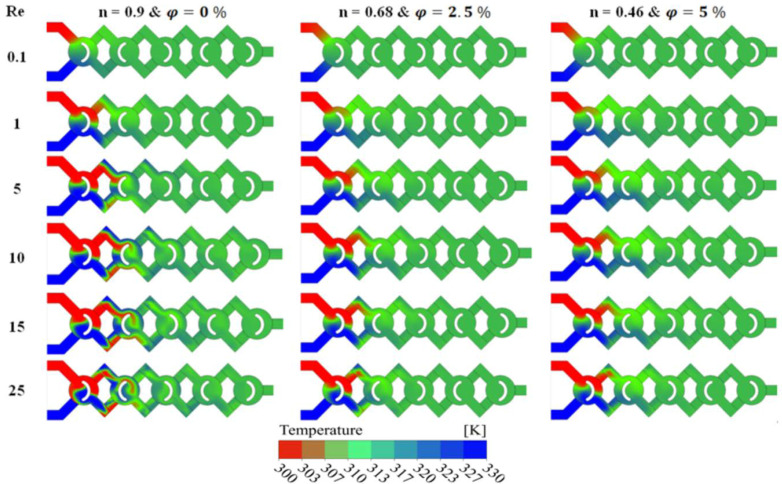
Qualitative representation of Temperature contours for different generalized Reynolds number at the horizontal middle section of each nano-fluid concentrations (φ = 0.5 to 5%).

**Figure 12 micromachines-13-00933-f012:**
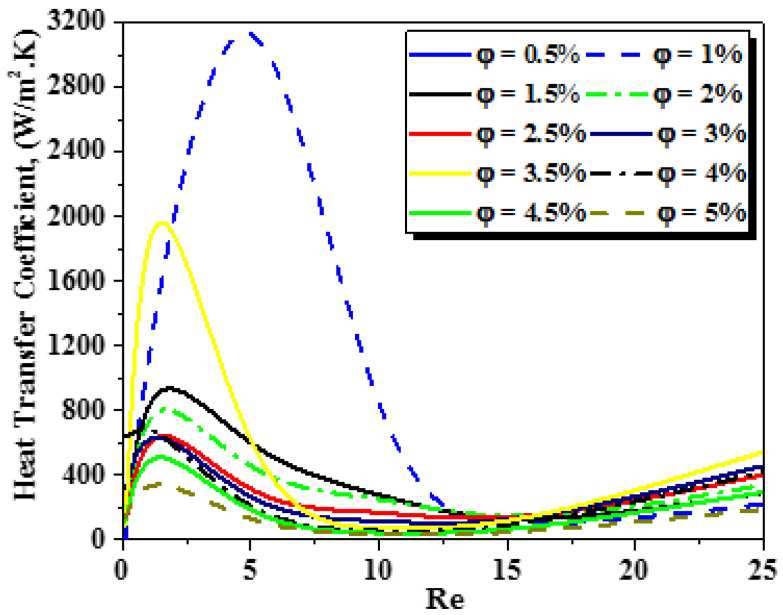
Evolutions of heat transfer coefficient for different generalized Reynolds numbers with various nanofluid concentrations (φ = 0.5 to 5%).

**Figure 13 micromachines-13-00933-f013:**
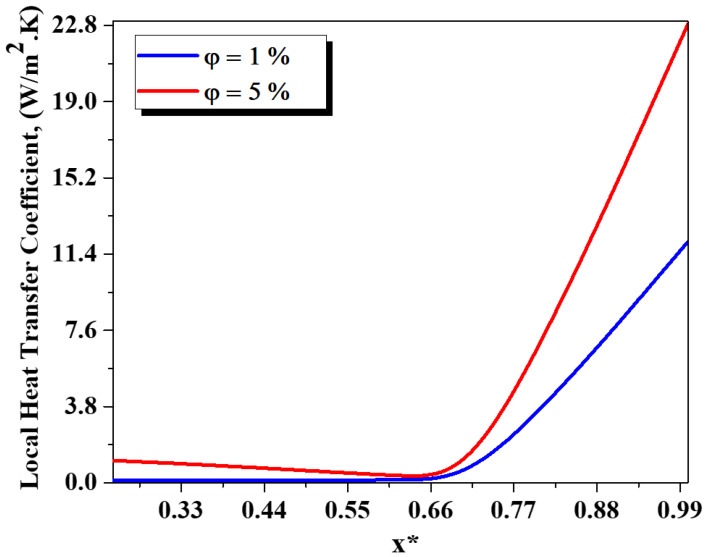
Evolutions of local heat transfer coefficient as a function along the geometry with φ = 1 and 5%.

**Figure 14 micromachines-13-00933-f014:**
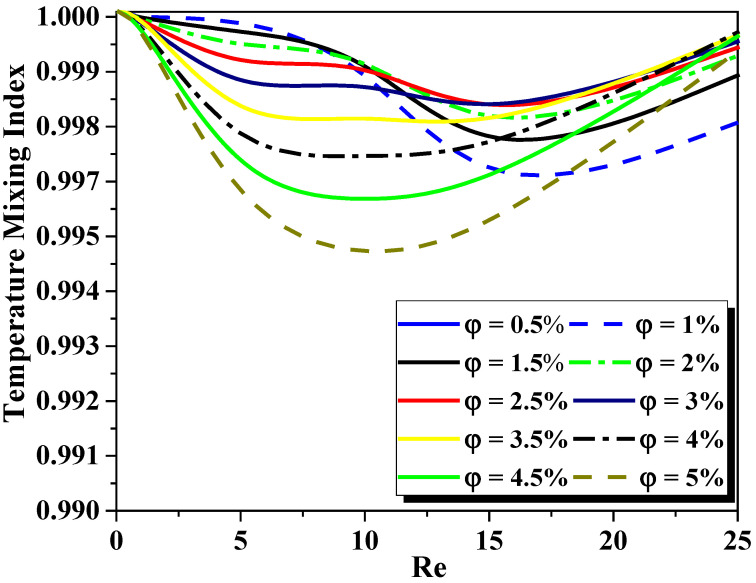
Improvement of thermal mixing performance for different generalized Reynolds numbers for variation cases of nanofluid concentrations (φ = 0.5 to 5%).

**Figure 15 micromachines-13-00933-f015:**
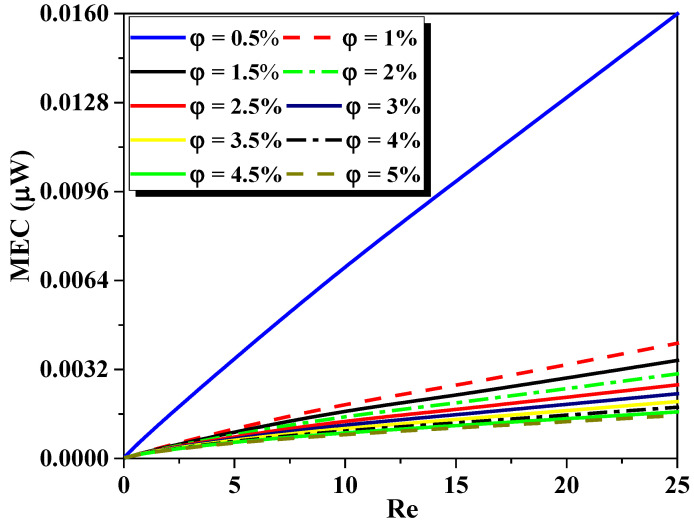
Development of Mixing Energy Cost for several generalized Reynolds numbers (Re_g_ = 0.1 to 25) for various nanofluid concentrations (φ = 0.5 to 5%).

**Table 1 micromachines-13-00933-t001:** Dimensions information of the proposed micromixer.

W	0.2 mm
*l**	0.8 mm
D*	0.2 mm
d*	0.1 mm
d_hyd_	0.22 mm
D	0.6 mm
L	4.5 mm

**Table 2 micromachines-13-00933-t002:** Rheological parameters of Al_2_O_3_ Nano-non-Newtonian, reproduced with permission from [23,24].

φ %	m (N s^n^ m^−2^)	N
0.5	0.00187	0.88
1.0	0.00230	0.83
1.5	0.00283	0.78
2.0	0.00347	0.730
2.5	0.00426	0.680
3.0	0.00535	0.625
3.5	0.00641	0.580
4.0	0.00750	0.540
4.5	0.00876	0.500
5.0	0.01020	0.460

**Table 3 micromachines-13-00933-t003:** Thermal parameters of Al_2_O_3_ Nano-non-Newtonian, reproduced with permission from [23,24].

(φ %)	ρ (Kg/m^3^)	$C_{p}\;(J/\mathrm{Kg}\;k)$	k (w/m k)
0.5	1013.1	4165.9	0.6248
1.0	1027.9	4148.8	0.6367
1.5	1042.8	4131.7	0.6488
2.0	1057.6	4114.6	0.6610
2.5	1072.5	4097.5	0.6734
3.0	1087.4	4080.5	0.6859
3.5	1102.2	4063.4	0.6987
4.0	1117.1	4046.3	0.7116
4.5	1131.9	4029.2	0.7246
5.0	1146.8	4012.1	0.7379

**Table 4 micromachines-13-00933-t004:** Local Temperature PDF ranging from 314 K to 316 K for different cross-sections.

φ (%)	P1 = 0.93 mm	P2 = 1.82 mm	P3 = 2.62 mm	P4 = 3.47 mm	Outlet
0.5	5%	31.45%	99.99%	99.99%	99.99%
2	7.83%	26.23%	99.99%	99.99%	99.99%
4	7.85%	24.43%	99.99%	99.99%	99.99%
5	12.24%	30.49%	99.99%	99.99%	99.99%

**Table 5 micromachines-13-00933-t005:** Quantitative comparison of the Energy Mixing Cost for varied generalized Reynolds number (Re = 1, 5, 15 and 15) with that obtained by Embarek et al. reprinted with permission from [41] in 2021y.

Re	MEC (μW) of Embarek [39]	MEC (μW) of Present Work	Evolution Parentage
1	0.00920	0.0064	30%
5	0.23600	0.194	18%
15	2.57302	1.548	40%
30	12.99	7.43	45%

**Table 6 micromachines-13-00933-t006:** Comparison of the pressure losses, mixing degree and energy mixing cost with different recent works at fixed Reynolds number (Re = 70).

Micromixer, Year	Unit	Length (mm)	ΔP (Pa)	MI	MEC (μW)
Split and recombination micromixer, 2012 [14]	6	10.66	16500	0.83	210.25
Convergent and divergent micromixer, 2017 [41]	6	11	14000	0.94	209.55
rectangular obstacles micromixer 2019 [45]	5	8.0	12542	0.98	179.75
triangular obstacles micromixer 2019 [45]	5	8.0	11485	0.93	175.38
teardrop obstacles micromixer 2019 [45]	5	8.0	10908	0.94	163.22
Elongation micromixer with 2021 [41]	4	3.5	8002	0.99	111.48
Present Two-layer Modified micromixer	5	4.58	3455	0.99	55.67

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
