# Peer review of "Evaluation of Hydrodynamic and Thermal Behaviour of Non-Newtonian-Nanofluid Mixing in a Chaotic Micromixer"

_micromachines, 2022, doi:10.3390/mi13060933_

Round 1

Reviewer 1 Report

The 3D CFD study on a novel micromixer seems interesting and with planty of applications in biomedical and chemical industrial fields. After my review, I have the following comments:

  1. Equations must be well writen. Use punctuation after equations throughout the manuscript.
  2. Adapt variables and equations in the text properly. See for instance line 126. The format is different in e.g. line 134.
  3. Al2O3 is a nanofluid. As such, an accurate simulation should be developed in Fluent as multiphase rather than non-Newtonian fluid. This is important to be justified and compared via two different simulations (multiphase vs non-Newtonian vs Experimental data) to justify and validate the simulation. 
  4. Where is the grid convergence study? In CFD, this is compulsory to trust sufficiently the simulation. 
  5. Did you validate the simulation with experimental data from other Al2O3-based works? This is important to trust the results. Validation with Xia et al [15] and Hossain et al [16] may not be  sufficient, unless a detailed grid convergence analysis justifies that the size of the mesh is appropriate for the transition between the comparison with Xia and Hossain and the novel application. In other words, the fluids are different and the geometry too, so please further justify why your results are reliable. Remember my previous point on multiphase vs nanofluid comparison, so that you could put everything into the same context.
  6. The image quality of several figures (e.g. Figure 3) is too bad. They are blurred. Please increase the quality of the images.
  7. The writing MUST be amended. Many many typos across the document, which reflect lack of proofreading.
  8. Table 6: Are you sure all those results can be compared? Fluids are different, thus to compare only with a fixed Reynolds number seems not practical to me, as other dimensionless quantites such as Peclet or Schmidt numbers differ. Please better justify your comparison.

In order to be accepted for publication, these points must be addressed. My current decision is major changes.

Reviewer 2 Report

The paper is interesting and worth’s publication. It is well organized and there is also comparison with other works. However, they should take into account some points into account and revise accordingly their manuscript.

Comment 1: In all the body of the manuscript there are typos errors exist. For example, Al2O3 should change to Al2O3. At lines 140-144 authors should choose between ( ) and [ ]. Other errors located in lines 164,167 etc.

Comment 2. Language error should be corrected for example

Line 310 “The development Mixing Energy Cost for various Reynolds number illustrates in table 5. For cases of Reynolds numbers” should be “The development Mixing Energy Cost for various Reynolds number is illustrated in table 5 for several cases of Reynold numbers”,

see also lines 119-120

The authors should correct have their manuscript corrected for such errors.

Comment 3: At line 47 authors claim that “Their proposed micromixer can give a fast mixing at minimum Reynolds numbers, for example, a high mixing index of 0.96 was found for very low flow regimes.” How fast? and what is the precise Reynolds number, definitely the authors are referring to Xia et all [15]. They should change to Their proposed micromixer can give a fast mixing (xxxx) at minimum Reynolds number (xxxx), for example, a high mixing index of 0.96 was found for very low flow regimes

Comment 4: At Figure 1, inside the periodic chamber the cavities run through the entire height of the micromixer and the sequency of cavities direction (left-right-left-…) was selected from previous work? Authors have tested another sequence such as left-left-right?

Comment 5: At line 197 authors claim that “Deviations of this order are not at all uncommon in numerical studies”. It is recommended to include some latest related research to support their claim. 

Numerical assessment of heat transfer and mixing quality of a hybrid nanofluid in a microchannel equipped with a dual mixer. International Journal of Thermofluids, 12, 100111 (2021)

Micromixing Nanoparticles and Contaminated Water Under Different Velocities for Optimum Heavy Metal Ions Adsorption Environ. Sci. Proc. 2020, 2(1), 65

Development of chaotic advection in laminar flow of a non-Newtonian nanofluid: a novel application for efficient use of energy. Applied Thermal Engineering, 124, 1213-1223 (2017)

 Mixing of Particles in Micromixers under Different Angles and Velocities of the Incoming Water,Proceedings 2018, 2(11), 577

Comment 6: At references section some of the references have different style.

Round 2

Reviewer 1 Report

The following points reflect how satisfied are my comments in the response by the reviewers:

Comment 1: Equations must be well written. Use punctuation after equations throughout the manuscript.

Mark: 0/100. The equations still have no punctuation.

Comment 2: Adapt variables and equations in the text properly. See for instance line 126. The format is different in e.g. line 134.

Mark: 100/100.

Comment 3: Al2O3 is a nanofluid. As such, an accurate simulation should be developed in Fluent as multiphase rather than non-Newtonian fluid. This is important to be justified and compared via two different simulations (multiphase vs non-Newtonian vs Experimental data) to justify and validate the simulation.

Mark: 50/100. I appreciate all the explanation but this is not the point. The point is that to simulate the behaviour of a nanofluid is not an easy task. It is well known that many researchers use single fluid properties, but the reality is that the fluid is not that monophase, especially at higher concentrations. Anyway, as I have seen validation with experimental data, I accept without major issues the use of a single phase approach. It is actually the best option to test geometries to improve mixing, but things must be validated first.

Comment 4:Where is the grid convergence study? In CFD, this is compulsory to trust sufficiently the simulation.

Mark: 100/100.

Comment 5:Did you validate the simulation with experimental data from other Al2O3-based works? This is important to trust the results. Validation with Xia et al [15] and Hossain et al [16] may not be sufficient, unless a detailed grid convergence analysis justifies that the size of the mesh is appropriate for the transition between the comparison with Xia and Hossain and the novel application. In other words, the fluids are different and the geometry too, so please further justify why your results are reliable. Remember my previous point on multiphase vs nanofluid comparison, so that you could put everything into the same context.

Mark: 70/100. Please translate the legend into English.

Comment 6:The image quality of several figures (e.g. Figure 3) is too bad. They are blurred. Please increase the quality of the images.

Mark: 60/100. Quality of images still vary between figures. Anyway, I believe the quality of the images is sufficient for publication.

Comment 7: The writing MUST be amended. Many many typos across the document, which reflect lack of proofreading.

Mark: 60/100. There are still some improvements over the text that should be addressed by a native speaker, but the writing is acceptable for publication to me. 

Comment 8: Table 6: Are you sure all those results can be compared? Fluids are different, thus to compare only with a fixed Reynolds number seems not practical to me, as other dimensionless quantites such as Peclet or Schmidt numbers differ. Please better justify your comparison.

Mark: 100/100.

Overall impression: 70/100. The proof of the manuscript is still a bit poor but sufficient to pass this review. I recommend the paper for publication once the unsatisfied comments are covered.

Reviewer 2 Report

The authors have responded appropriately to the points raised by the reviewers. Thus I recommend it for publication. Just a minor comment is tocheck if sirnames appear correctly in the added references 35, 46.